# Sex Difference in Cardioprotection against Acute Myocardial Infarction in MAO-B Knockout Mice In Vivo

**DOI:** 10.3390/ijms24076443

**Published:** 2023-03-29

**Authors:** Jacqueline Heger, Tamara Szabados, Paulin Brosinsky, Péter Bencsik, Péter Ferdinandy, Rainer Schulz

**Affiliations:** 1Institute of Physiology, Justus Liebig University, 35392 Giessen, Germany; 2Department of Pharmacology and Pharmacotherapy, University of Szeged, 6722 Szeged, Hungary; 3Pharmahungary Group, 6722 Szeged, Hungary; 4Department of Pharmacology and Pharmacotherapy, Semmelweis University, 1094 Budapest, Hungary

**Keywords:** monoamine oxidase B, mitochondria, ischemia/reperfusion

## Abstract

The cardiomyocyte-specific knockout (KO) of monoamine oxidase (MAO)-B, an enzyme involved in the formation of reactive oxygen species (ROS), reduced myocardial ischemia/reperfusion (I/R) injury in vitro. Because sex hormones have a strong impact on MAO metabolic pathways, we analyzed the myocardial infarct size (IS) following I/R in female and male MAO-B KO mice in vivo. Method and Results: To induce the deletion of MAO-B, MAO-B KO mice (Myh6 Cre+/MAO-B^fl/fl^) and wild-type (WT, Cre-negative MAO-B^fl/fl^ littermates) were fed with tamoxifen for 2 weeks followed by 10 weeks of normal mice chow. Myocardial infarction (assessed by TTC staining and expressed as a percentage of the area at risk as determined by Evans blue staining)) was induced by 45 min coronary occlusion followed by 120 min of reperfusion. Results: The mortality following I/R was higher in male compared to female mice, with the lowest mortality found in MAO-B KO female mice. IS was significantly higher in male WT mice compared to female WT mice. MAO-B KO reduced IS in male mice but had no further impact on IS in female MAO-B KO mice. Interestingly, there was no difference in the plasma estradiol levels among the groups. Conclusion: The cardiomyocyte-specific knockout of MAO-B protects male mice against acute myocardial infarction but had no effect on the infarct size in female mice.

## 1. Introduction

Monoamine oxidase (MAO) is a protein located in the outer mitochondrial membrane [1], where it degrades neurotransmitters and biogenic amines. The oxidative deamination of primary, secondary and tertiary amines is coupled to the reduction in the covalently linked co-factor flavin adenine dinucleotide (FAD) [2]. With the aid of water and oxygen, the formation of aldehydes, ammonia and the reactive oxygen species (ROS) molecule hydrogen peroxide (H_2_O_2_) are formed in the mitochondrial matrix and in the cytosol of the cell [3,4,5]. FAD is thereby reduced to FADH_2_; it accepts two hydrogen atoms and undergoes a net gain of two electrons. The recycling of FAD generates H_2_O_2_. The interaction of the FAD side chain with MAO-B is of utmost importance to incorporate FAD into MAO-B [6], because a deformation in the flavin ring negatively affects the reactive center and plays a crucial role for irreversible MAO inhibitor binding [7]. An accumulation of catecholamines can result in numerous diseases, such as Brunner syndrome [8], arrhythmia or general cardiovascular problems [9,10]. There are two isoforms of MAO, MAO-A and MAO-B, which are both strongly expressed in the heart [11,12]. They differ in their substrate specificity and inhibitor sensitivity. MAO-A primarily oxidizes serotonin and norepinephrine, while both MAO-A and MAO-B metabolize tyramine and dopamine. β-phenylethylamine (PEA) is a biogenic trace amine and a substrate for MAO-B [13,14].

Myocardial infarction (MI) occurs as a consequence of the prolonged interruption of the blood supply to the heart. If patients survive a severe MI, the heart function deteriorates, and heart failure (HI) often develops. While reperfusion through primary percutaneous coronary intervention is the only possibility to reduce myocardial infarct size, reperfusion itself induces further damage to the heart muscle, known as reperfusion injury [15,16].

Oxidative stress is a major contributor to irreversible ischemia/reperfusion (I/R) injury. Already during ischemia, the amount of ROS rises and increases further with the beginning of reperfusion. Apart from enzymes in the cytosol of the cell, ROS molecules are also generated by mitochondria [17]. Under stress conditions, such as I/R, the autonomic nervous system is activated, releasing neurotransmitters such as norepinephrine, which in turn are broken down by MAOs leading to the formation of H_2_O_2_, which can directly influence heart function [18].

Besides norepinephrine, serotonin (5-HT) and histamine also play important roles in I/R injury. 5-HT accumulates in the heart during ischemia and is degraded after reperfusion depending on the MAO-A activity after uptake into the cells [19]. Histamine co-localizes with norepinephrine in neurons [20] and is enclosed in cytoplasmatic granules of mast cells, which lie adjacent to blood vessels and between cardiomyocytes [21]. Mast cells become activated by I/R and release histamine [22] and histamine released from the heart is increased during I/R [21,23,24]. Histamine is metabolized to 1-N-Methylhistamine (1N-Met) which acts as a substrate for MAO-B, thereby increasing the ROS formation [24].

The MAO pattern in different cell types has been examined by the use of specific inhibitors, where MAO-A is inhibited by low concentrations of clorgyline and MAO-B by selegiline. Both isoforms are inhibited by pargyline [25]. MAOs play an essential role in the nervous system by modulating the levels of neurotransmitters. Various neurological diseases such as depression, Alzheimer’s and Parkinson’s are associated with altered MAO levels. The different preferences in substrate affinity are crucial for the different clinical significance of the two MAOs [26]. The inhibition of both isoforms or the use of irreversible MOA inhibitors can be fatal by the accumulation of MAO substrates, such as tyramine or serotonin. Increased amounts of tyramine can enter the systemic circulation and, from there, adrenergic neurons, consequently increasing the noradrenaline release and resulting in a severe hypertensive response [5,27]. Similarly, an increase in serotonin concentration might be deleterious, leading to serious adverse outcomes, including death [28]. Selective MAO-B inhibition might provide a cardiac benefit while MAO-A can maintain tyramine and serotonin homeostasis.

Besides experiments with MAO-specific inhibitors [29], MAO activity is inhibited by gene targeting in mice [13,30,31]. MAO-B inhibition prevented the doxorubicin-induced cardiac dysfunction [29]. We have previously shown that the cardiomyocyte-specific deletion of MAO-B is accompanied by a reduction in mitochondrial ROS production upon administration of specific substrates and infarct size and accelerated functional recovery following I/R in Landendorff-perfused male and female mice hearts in vitro [32].

Gender differences in cardiac I/R injury have already been described (for a review, see [33,34,35,36]. Such an increased resistance of female hearts to I/R in vivo and in vitro was reported for several species [33,37,38] and has been attributed to the potential cardioprotective mechanism(s) of estrogen [33]. Interestingly, females have higher plasma MAO activity than males [39] and estrogens can modulate MAO activity [40]. Women show a different susceptibility to diseases than men and react differently to treatment. The incidence of depression in women is nearly double that in men. Female susceptibility may be linked to surges in reproductive hormones [41]. Women had a statistically superior response to MAOIs as antidepressant treatment [42], demonstrated by elevated levels of serotonin, as well as the serotonin metabolite 5-hydroxyindoleacetic acid (5-HIAA), to be found in women compared to men [41]. This makes a study of females and males with MAO-B KO even more promising.

In the present study, we therefore wanted to 1. confirm our previous in vitro findings on the cardiomyocyte-specific deletion of MAO-B being cardioprotective in an in vivo mice model of I/R injury and 2. compare the cardioprotective potential in female and male hearts.

## 2. Results

### 2.1. Cardiomyocyte-Specific MAO-B Deletion and Mortality Rate after I/R

There was no major difference in the mortality rate between the wild-type (WT) and cardiomyocyte-specific MAO-B knockout (KO) females or males, as well as in the WT females vs. KO females. However, comparing all the male to female mice, there was a tendency for a higher mortality rate in the males compared to the females (*p* = 0.103, when analyzed by Fisher’s exact test) (Figure 1).

### 2.2. Cardiomyocyte-Specific MAO-B Deletion and Infarct Size

The WT male mice had a significantly higher infarct size than the WT female mice following 45 min ischemia and 120 min reperfusion. There was no significant difference in the area at risk between the different groups (Figure 2A). The cardiomyocyte-specific knockout of MAO-B (KO) reduced the myocardial infarct size in the male hearts but did not affect the infarct size in the female hearts (Figure 2B). Interestingly, the infarct size in the male KO was similar to that of both the WT and KO female hearts following I/R.

### 2.3. Cardiomyocyte-Specific MAO-B Deletion and Heart Rate

After reperfusion, the female WT as well as the male and female KO mice responded with a significant increase in heart rate. Only the WT male hearts did not show an increase in heart rate (Figure 3).

In addition, the differences in the number of reperfusion arrhythmias were measured among the different groups (Figure 4). The incidence and severity of the arrhythmias were evaluated according to convention 23 of the Lambeth conventions II [43]. The male mice displayed more severe arrhythmias than the females as evaluated by the arrhythmia score. The arrhythmia score was calculated according to the phenotype and duration of the arrhythmias. The ventricular premature beats (1), ventricular bigeminy (2) and ventricular salvos (3) were scored; however, neither ventricular tachycardia nor fibrillation occurred in any of the individuals. The duration of the arrhythmias was expressed in minutes. In the cases where several occurrences of different types of arrhythmias were found in one individual, the most severe arrhythmia with the longest duration was used for the analysis. The data are expressed as a violin plot and analyzed by the Kruskal–Wallis test (four groups) as well as by Mann–Whitney non-parametric tests (when the difference between the males and females was tested). The Mann–Whitney test resulted in only a decreasing tendency in the WT female mice as compared to the KO male mice (*p* = 0.054). However, when the WT and KO males as well as the WT and KO females were united, the Mann–Whitney test showed a significantly decreased arrhythmia score in the female mice as compared to the males (*p* = 0.014), regardless of gene modification.

### 2.4. Cardiomyocyte-Specific MAO-B Knockout and Heart Weight

The female mice had reduced heart weights (HW) and body weights (BW) compared to the male mice (Figure 5A,B). The HW/BW ratios were similar in the WT males and females (Figure 5C) as well as in the male cardiomyocyte-specific knockout (KO) mice, while in the female KO mice the HW/BW ratio was reduced compared to the male mice.

### 2.5. Cardiomyocyte-Specific MAO-B Knockout and 17β Estradiol

As estradiol might affect I/R injury and MAO activity, 17β estradiol levels were measured (Table 1). The male MAO-B KO had similar estradiol levels as the WT as well as the KO females. In contrast, the WT males had significantly higher estradiol levels compared to all other groups.

## 3. Discussion

Female WT are more resistant to I/R injury and develop a smaller infarct size after I/R compared to male WT mice. The cardiomyocyte-specific knockout of MAO-B reduced IS in male mice but had no (additive) protective effect in female mice. Protection by estrogen may be due to altered gene expression.

Sex difference in I/R injury: Estradiol can protect the heart against ischemia-induced injury by blocking the mitochondrial permeability transition pore (MPTP) opening [44]. Moreover, estrogens can preserve cardiac function after I/R by diminishing cytokine levels [45]. In women with estrogen therapy because of depression, estradiol decreases the expression of MAO-A and MAO-B through the interaction with its intracellular receptors [46]. Estradiol binds to different estrogen receptors (ER) (ER-α and ER-β) or a G-protein-coupled estrogen receptor (GPR30 or GPER) [47], which can thereby regulate various physiological and pathological processes [48]. ER-β localizes to mitochondria, thereby modulating the mitochondrial calcium influx, ATP production, apoptosis and free radical species formation, all processes critically involved in I/R injury [49] (for a review, see [36]).

Thus, although estradiol is important for I/R injury, there was no evidence of an increase in circulating plasma 17β-estradiol concentrations and cardioprotection in the different groups of mice in the present study. It is possible that the timing of the sampling is not appropriate or that 17β estradiol has already been internalized in a receptor-bound manner. In addition, Klaiber et al. [50] reported that mean values of serum estradiol levels in men with myocardial infarction were significantly elevated over the comparable mean values of a respective control group. Estrogens can increase adrenergic activity, which would increase myocardial oxygen demand, and testosterone can be converted to estradiol by aromatase in muscles.

Testosterone increases the gene expression of enzymes such as MAO through the direct activation of androgen receptors (AR) [51]. During I/R, AR increase in males as compared to females, leading to increased cardiac injury via the activation of apoptosis [52]. We did not measure the testosterone levels or AR in the present study, but in our previous study [32], the infarct size did not differ between male and female hearts, making such an explanation unlikely.

Apart from sex hormones, differences are found in the regulation of the cell survival pathways in males and females [16]. Cardioprotection in female rats after I/R is mediated by altered mitochondrial enzyme activity that encompasses a phosphoinositide-3-kinase (PI3K)-mediated reduction in ROS generation and a better removal of ROS by-products [37]. Interestingly, females have increased activity of aldehyde dehydrogenase 2 (ALDH2) [37]. MAOs degrade neurotransmitters and biogenic amines which, in addition to the production of ROS, also leads to the formation of aldehydes [3,4]. ALDH2 inactivates aldehydes in the heart [53] and ALDH2 activation before an ischemic event decreased the infarct size [54]. ALDH2 is located in the mitochondria and is important for the detoxification of 4-hydroxy-2-nonenal (4-HNE), which is produced during oxidative stress [55,56]. The downregulation of ALDH2 in cardiomyocytes by siRNA impaired the mitochondrial function demonstrated by a reduction in the mitochondrial membrane potential [57]. The overexpression of ALDH2 protects against 4-HNE accumulation and cardiac dysfunction in transgenic mice with the cardiac overexpression of MAO-A [56]. However, the aldehyde intermediates generated by MAO-B may also contribute to the alteration of the mitochondrial function [57]. Thus, an altered expression of ALDH2 could be important for the gender-specific cardioprotection against I/R. A possible approach to investigate the importance of gender, MAO-B and ALDH2 in the context of I/R injury would be the use of a MAO-B-specific inhibitor in ALDH2-overexpressing mice [58]. The use of MAO inhibitors could lead to an additive effect, thereby confirming the gender difference. If the cardioprotective effect in females is due to the increased activity of ALDH2, no difference should be seen between ALDH2-overexpressing mice with and without MAO inhibition. However, these experiments would completely go beyond the scope of this work.

Mortality and infarct size: It is well established that an increased infarct size is associated with increased mortality. Thus, the reduced mortality in WT female mice as compared to WT male mice is in accordance with this assumption. However, while the infarct size was reduced by the cardiomyocyte-specific knockout of MAO-B in male mice, the mortality rate remained high. Although the extent of reperfusion-induced arrhythmias was reduced, the latter potentially being explained by the desensitization of β-adrenoceptors. Wang et al. [59] discovered that MAOs desensitize the β_1_-adrenergic pathway in mice injected intraperitoneally with the β-agonist isoproterenol to induce heart failure (HF). The inhibition of MAO-A in isolated adult cardiomyocytes of HF hearts salvaged the signaling pathway and led to the activation of the protein kinase A [60]. Thus, more studies are needed to explain the differences in the mortality rate between cardiomyocyte-specific MAO-B knockout male and female mice following I/R.

Overall, it can be concluded that a cardiomyocyte-specific reduction in MAO-B contributes to cardioprotection in male but not in female mice. MAOs can be regulated by sex, but a detailed analysis of sex hormones influences on cell survival signaling pathways in a MAO-dependent manner remains to be elucidated.

## 4. Materials and Methods

### 4.1. Animals

All animals were housed in individually ventilated cages (Sealsafe IVC system, Tecniplast S.p.a., Buguggiate, Italy), which conform to the size recommendations in the most recent Guide for the Care and Use of Laboratory Animals DHEW (NIH Publication No. 85–23, revised 1996) and EU Guidelines 63/2010. Litter material placed beneath the cage was changed at least once a week. The animal room was temperature controlled having a 12-h light/dark cycle with lights on at 7 a.m. to 7 p.m., and was kept clean and vermin free. All animal experiments conformed to the EU directive about the care and use of laboratory animals, published by the European Union (2010/63/EU), and it was approved by the National Scientific Ethical Committee on Animal Experimentation (approval ID: XXVIII./171/2018.; on 24 January 2018). Generation of cardiac-specific and tamoxifen-inducible double-transgenic MAO-B KO mice (Myh6-MCreM_x_MAO-B^fl/fl^) was described previously [33]. Preliminary sample size was calculated (https//:www.sample-size.net, accessed on 22 May 2020). The values were set up in case of females to α = 0.22 (type I error rate); β = 0.44 (type II error rate); P_0_ = 0.15 (Group 0 risk); P_1_ = 0.4 (Group 1 risk); sample size = 12. The values were set up in case of males to α = 0.22 (type I error rate); β = 0.44 (type II error rate); P_0_ = 0.2 (Group 0 risk); P_1_ = 0.38 (Group 1 risk); sample size = 24.

### 4.2. Genotyping

Ear biopsies were digested in 150 µL “Direct PCR Tail reagent” (#31-101-T, Peqlab [VWR, Darmstadt, Germany]) supplemented with 3 µL Proteinase K (20 mg/mL) and 150 µL H_2_O at 55 °C for seven hours. Afterward, samples were incubated for 45 min at 85 °C. Samples were shortly centrifuged and supernatant was temporarily stored at 4 °C for subsequent PCR. Homozygous, heterozygous and wild-type mice were identified using the following primers:
MAO-B^fl/fl^ forward5′-GCC CAC GAG TAA GTA AAT ACG TGG A-3′MAO-B^fl/fl^ reverse5′ GGT CTC TGT TTC TGG GAC AGT CTG-3Myh6-MCreM forward5′-GAC CAG GTT CGT TCA CTC ATG G-3′Myh6-MCreM revererse5′-AGG CTA AGT GCC TTC TCT ACA C-3′

PCRs were performed with Taq-polymerase-kit (#10342046, Invitrogen [Thermo Fisher Scientific, Miami, OK, USA]) with a MyCycler thermal cycler (BioRad, Feldkirchen, Germany). For Myh6-MCreM, annealing temperatures of 55 °C, 35 cycles were used, and for MAO- B^fl/fl^, 62 °C, 34 cycles were used. Bands were detected with Gelred (Biotium, Fremont, CA, USA) in a 2% agarose gel after 45 min run in 1x TBE. Amplification of genomic DNA resulted in PCR products of either 226 bp or 353 bp. Presence of both products identifies heterozygous mice, 353 bp product homozygous mice and 226 bp product wild-type mice.

### 4.3. Determination of Infarct Size In Vivo

Briefly, Evans blue was injected in vivo into the apex of the left ventricle after re-occlusion of the coronary artery. Evans blue colors the whole left ventricle excluding the area supplied by the occluded coronary artery to dark blue, thereby demarcating the area at risk. In the next step, the heart is excised and cut into 5 to 6 slices and incubated in vitro with 1% TTC at 37 °C. TTC binds to the reduced coenzymes such as NADH and FADH2 produced by the surviving cells, which leads to the conversion of TTC to a formazan dye and changes its color to red. Red-colored area is then identified as the surviving area inside the area at risk. Unstained pale areas are identified as infarct size. In detail: The mice were anesthetized by intraperitoneal (90 mg/kg i.p.) injection of sodium pentobarbital (Repose 50%, Le Vet. Pharma, Oudewater, The Netherlands) and were mechanically ventilated via tracheal cannula using room air (Model 845, Minivent, Harvard Apparatus, Holliston, MA, USA) in a volume of 150–200 µL and frequency of 140–150 strokes/min according to the recommendation of the ventilator’s manufacturer. A left parasternal thoracotomy was performed, and the intercostal muscles were opened in the region of the 4th intercostal space. The pericardium was opened, and the left descending coronary artery (LAD) was occluded at its middle portion using an 8–0 synthetic, monofilament, non-absorbable polypropylene suture (Prolene^®^, Ethicon, Johnson & Johnson Kft Hungary, Budapest, Hungary) and a 3 cm piece of PE50 cannula to form a snare. For coronary artery occlusion, the plastic tubing was pressed onto the surface of the heart directly above the coronary artery and released for reperfusion. Myocardial infarction was induced by 45 min coronary occlusion followed by 120 min of reperfusion. Surface-lead ECG (Haemosys, Experimetria Inc., Budapest, Hungary) and body core temperature were monitored throughout the experiments to ensure the stability of the preparation. The successful induction of ischemia was visible as a pale coloring of the myocardium as well as changes in the ECG (ST-elevation, broadening of the QRS complex). At the end of the 120 min reperfusion, coronary artery was re-occluded, a maximum of 0.5 mL blood was collected by apical puncture of the left ventricle and gentle suction into a 1 mL syringe and immediately into Li-heparin tubes for plasma separation. Blood samples were centrifuged at 1000× *g* for 15 min at 4 °C, plasma was separated and immersed immediately into liquid nitrogen and stored at −70 °C until biochemical analysis. The heart was excised from the chest, the occlusion was released and atria and right ventricle were removed. The left ventricle was quickly frozen at −20 °C for 10 min and then cut into 1 mm thick slices. Infarct size was determined by standard 2,3,5-triphenyltetrazolium chloride (TTC) staining (1% for 10 min at 37 °C). Stained heart slices were put between two thin sheets of glass, and then photographed using a digital zoom camera (Canon Powershot SX60HS, Canon, Tokyo, Japan). Digital heart images were processed (both surfaces of the slices were merged into one image) and infarct size was evaluated by planimetry using InfarctSize™ v.2.5 software (developed by Pharmahungary, Budapest, Hungary) and expressed as a percentage of the area at risk [61].

### 4.4. Determination of 17 β Estradiol

Plasma samples were taken at the end of reperfusion period (at 120 min) just before intracardiac injection of Evans blue. The 17 β estradiol measurements were performed by using 17 beta Estradiol ELISA Kit (ab108667) of abcam (Cambridge, UK). The test was performed according to the manufacturer’s instructions using plasma samples of mice after I/R.

### 4.5. Statistical Analysis

Statistical analysis was performed using SPSS (IBM SPSS Statistics 29). Results are represented as box plots and expressing median, 25% and 75% quartiles, upper and lower whisker and outliers as well as a violin plot for displaying reperfusion arrhythmias. The difference between two groups was compared using a two-tailed unpaired *t*-test. Comparison between more than two groups was performed with two-way ANOVA.

A one-way ANOVA, Kruskal–Wallis, Mann–Whitney and Student’s *t*-test were used for the further evaluation of differences between two means. Values of *p* < 0.05 were considered to be statistically significant.

## Figures and Tables

**Figure 1 ijms-24-06443-f001:**
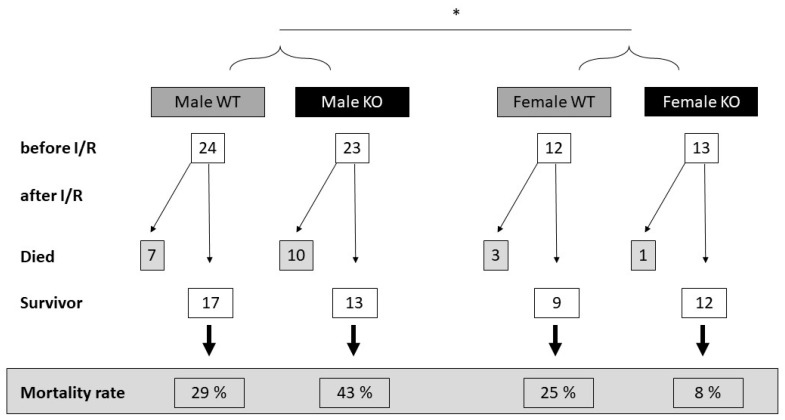
Higher mortality rate in male wild-type (WT) and cardiomyocyte-specific MAO-B knockout (KO) mice compared to female mice. WT and KO hearts were exposed to 45 min ischemia and 120 min reperfusion. Summary of the number of operated mice before and after surgery. Excluded mice had blood in the urine or surgery complications. There is no significant difference among groups when analyzed groups separately by Chi-square test or between WT and KO animals when analyzed by Fisher’s exact test. However, there is a slightly decreasing tendency but not significant in all-cause mortality in female individuals as compared to males when analyzed by Fisher’s exact test (* *p* = 0.103).

**Figure 2 ijms-24-06443-f002:**
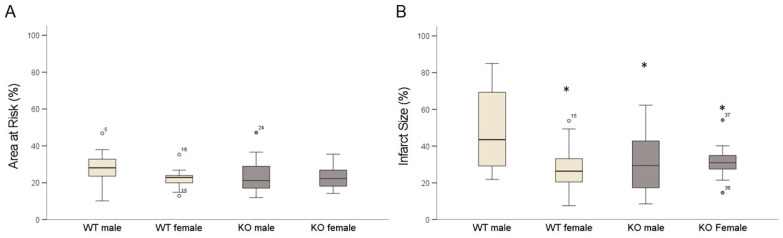
Male MAO-B KO mice benefit from MAO-B deletion regarding reduced size of infarcted area. (**A**) shows quantification of area at risk and (**B**) infarct size quantification. (WT male n = 14; WT female n = 9; KO male n = 11; KO female n = 11). Data are represented as box plots expressing median, 25% and 75% quartiles, upper and lower whisker and outliers. * *p* < 0.05 WT male vs. WT female, KO male, KO female, Student’s *t*-test.

**Figure 3 ijms-24-06443-f003:**
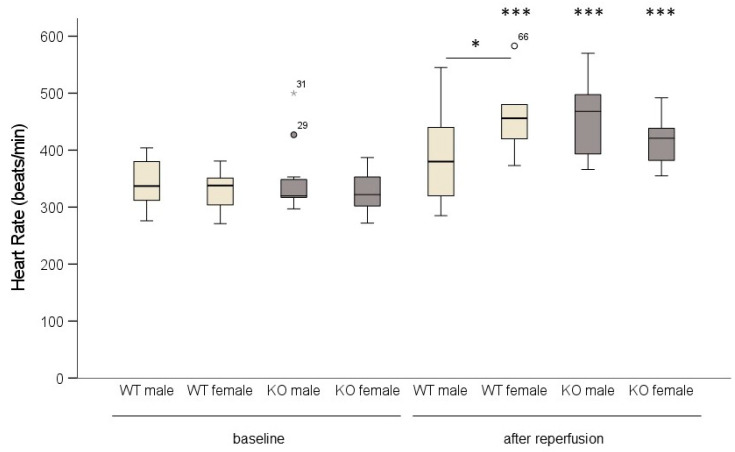
In male wild-type (WT) mice, heart rate did not increase following ischemia/reperfusion. WT and cardiomyocyte-specific MAO-B knockout (KO) hearts were exposed to 45 min ischemia and 120 min reperfusion. Heart rate was measured baseline and after reperfusion. (WT male n = 14; WT female n = 9; KO male n = 11; KO female n = 11). Data are represented as box plots expressing median, 25% and 75 % quartiles, upper and lower whisker and outliers (^○^,

). * *p* < 0.05, *** *p* < 0.001 baseline vs. after reperfusion, Student’s *t*-test. Two-way ANOVA: *p* < 0.001, baseline vs. after reperfusion *p* < 0.001, genotype * sex *p* = 0.02.

**Figure 4 ijms-24-06443-f004:**
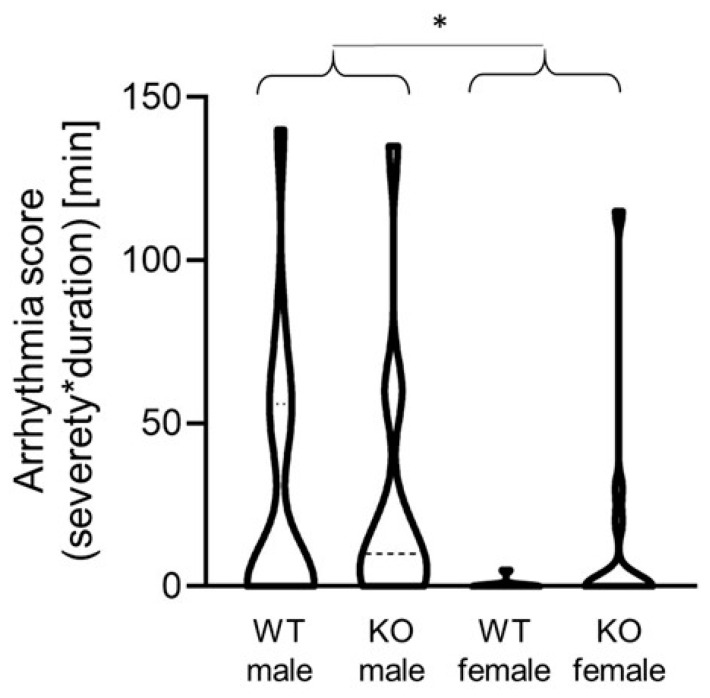
Violin plot representing the severity and duration of reperfusion arrhythmias. (WT male, male wild-type mice; KO male, male MAO-B knockout mice; WT female, female wild-type mice; KO female, female MAO-B knockout mice). * *p* = 0.014 when WT and KO males are compared to WT and KO females as analyzed by Mann–Whitney test.

**Figure 5 ijms-24-06443-f005:**
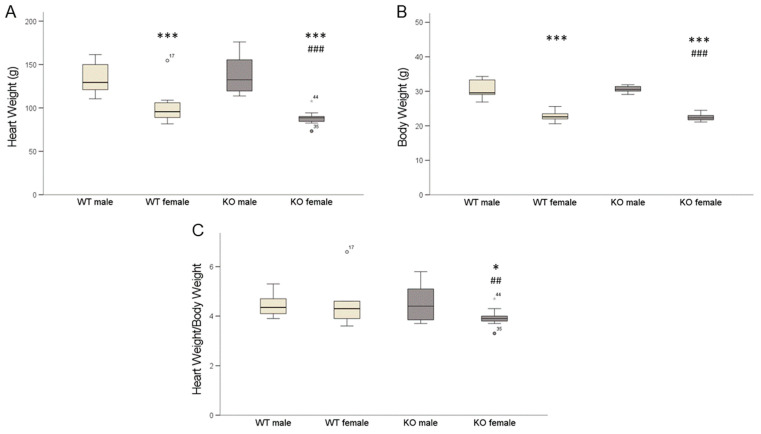
Heart weight (HW) and body weight (BW) as well as HW/BW ratios of mice. Female mice had reduced heart weight (**A**), *** *p* < 0.001 male vs. female and ^###^ *p* < 0.001 WT male vs. KO female, Student’s *t*-test; two-way ANOVA: sex *p* < 0.001. Female mice had reduced body weight (**B**) compared to male mice, *** *p* < 0.001 male vs. female, ^###^ *p* < 0.001 WT male vs. KO female, Student’s *t*-test; two-way ANOVA: sex *p* < 0.001. HW/BW ratios (**C**) were similar in WT males and females as well as in male cardiomyocyte-specific knockout (KO) mice, while in female KO mice HW/BW ratio was reduced, * *p* < 0.05 KO male vs. KO female and ^##^ *p* < 0.01 WT male vs. KO female, Student’s *t*-test. Two-way ANOVA: sex * genotype *p* = 0.088. (WT male n = 14; WT female n = 9; KO male n = 11; KO female n = 11). Data are represented as box plots expressing median, 25% and 75 % quartiles, upper and lower whisker and outliers (^○^,

).

**Table 1 ijms-24-06443-t001:** Plasma 17β estradiol levels in WT and MAO-B KO mice. WT and MAO-B KO hearts were exposed to ischemia for 45 min and reperfusion for 120 min. Plasma samples were collected and 17β estradiol levels determined. * *p* < 0.05 WT male vs. WT female, ^##^ *p* < 0.01 WT male vs. MAO-B KO male and female, Student’s *t*-test. Two-way ANOVA: genotype *p* = 0.005; sex *p* = 0.161.

	WT		MAO-B KO	
	Male	Female	Male	Female
Mean	27.9 *^,##^	25.1	23.7	23.9
SEM	0.81	0.96	1.01	0.81
N	9	9	8	11

## Data Availability

The data presented in this study are available on request from the corresponding author.

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
