# Peer review of "Sex Difference in Cardioprotection against Acute Myocardial Infarction in MAO-B Knockout Mice In Vivo"

_ijms, 2023, doi:10.3390/ijms24076443_

Round 1

Reviewer 1 Report

Summary:Since, Mono amino oxidase is strongly influenced by the sex hormones, the authors Heger et al., analyzed myocardial infarct size (IS) following I/R in female and male MAO-B KO mice in vivo. Myocardial infarction (assessed by TTC-staining and expressed as percentage of area at risk as determined by evens blue staining)) was induced by 45 min coronar occlusion followed by 120 minutes of reperfusion. The authors are concluding that MAO-B deletion protects male mice against acute myocardial infarction, as they found that MAO-B KO reduced IS in male mice but had no further impact on IS in female MAO-B KO mice.

Comments:

1.     Introduction is very clear and has enough background to understand about MAO.

2.     Can the authors explain the relation between the heart weight and MAO-B deletion? Does it pose a positive or negative effect on the overall Myocardial infraction condition?

Author Response

Reviewer 1

Comments and Suggestions for Authors

We thank the reviewer for her/his time and constructive questions to improve our manuscript. We have addressed the comments and have revised the manuscript accordingly.

Comments:

  1. Introduction is very clear and has enough background to understand about MAO.
  2. Can the authors explain the relation between the heart weight and MAO-B deletion? Does it pose a positive or negative effect on the overall Myocardial infraction condition?

So far, we cannot explain exactly why the HW/BW ratio seems to be somewhat reduced in female MAO-B KO mice after I/R compared to male mice (p<0.05 between female MAO-B KO mice and male MAO-B KO mice and p<0.01 between female MAO-B KO mice and male WT mice; two Way ANOVA sex versus genotype p=0.088). Echocardiographic studies of MAO-B KO mice revealed no changes in HW/BW ratio between males and females under basal conditions (data derived from another project).

Cardiac-specific deletion of MAO-B did not result in any change in body or heart weight in the present study. Rather, sex-specific differences between males and females are observed, i.e. females have smaller values for both body weight as well as heart weight; however, this sex difference is present in both wild type and MAO-B KO animals. Monoamine neurotransmitter like serotonin or dopamine, belong to the feeding-inhibitory molecules thereby affecting food intake (PMID: 10699154), which leads to decreased body weight. We found a previous paper, which reported reduced body weight of double MAO-A and MAO-B KO male mice which have increased PEA, serotonin and dopamine values (PMID: 15272015), however, heart weight and myocardial infarct size were not measured in that study. Although, we did not investigate the correlation between heart weight and infarct size, we suppose that body weight does not significantly affect the size of necrotic area in this mouse model of acute myocardial infarction.

Since weight was determined after I/R, differences are probably due to better compensation of remodeling. That an attenuated increase in heart weight can be protective was shown for cardiomyocyte-specific overexpression of estrogen receptor β in mice. In response to MI, these mice exhibited improved survival compared to the WT counterparts due to reduced heart weight (PMID: 26608078).

A correlation between MAO and estradiol has been shown previously:  Chaplain et al. (PMID: 4234068) observed in rats, that cardiac hypertrophy from experimental hypertension (sodium and other stimuli) was associated with a marked increase in MAO concentration. Conversely, a delayed heart growth during sodium restriction was accompanied by a lower MAO activity. The authors speculated that these changes in MAO seemed to be closely related to the mass of the heart. This phenomenon is also found in the MAO-B KO females, where less MAO correlates with a lower weight. They further state that a different distribution of MAO in the heart indicates that the localization of enzyme can vary between various organs and that this may reflect different metabolic functions.

To answer the question raised by the reviewer:  The heart weight alone is most likely not the reason for the observed protection, but the mechanism that led to the weight change, might also affect infarct development.

Inui A. Transgenic approach to the study of body weight regulation. Pharmacol Rev. 2000 Mar;52(1):35-61. PMID: 10699154.

Chen K, Holschneider DP, Wu W, Rebrin I, Shih JC. A spontaneous point mutation produces monoamine oxidase A/B knock-out mice with greatly elevated monoamines and anxiety-like behavior. J Biol Chem. 2004 Sep 17;279(38):39645-52. doi: 10.1074/jbc.M405550200. Epub 2004 Jul 22. PMID: 15272015; PMCID: PMC2861902.

Schuster I, Mahmoodzadeh S, Dworatzek E, Jaisser F, Messaoudi S, Morano I, Regitz-Zagrosek V. Cardiomyocyte-specific overexpression of oestrogen receptor β improves survival and cardiac function after myocardial infarction in female and male mice. Clin Sci (Lond). 2016 Mar;130(5):365-76. doi: 10.1042/CS20150609. Epub 2015 Nov 25. PMID: 26608078.

de Chaplain J, Krakoff LR, Axelrod J. Increased monomine oxidase activity during the development of cardiac hypertrophy in the rat. Circ Res. 1968 Sep;23(3):361-9. doi: 10.1161/01.res.23.3.361. PMID: 4234068.

Reviewer 2 Report

The manuscript by Jacqueline Heger et al. investigate how gender and lack of monoamine oxidase (MAO)-B, an enzyme involved in the production of reactive oxygen species (ROS), may influence myocardial infarct size following ischemia/reperfusion (I/R) injury in an in vivo experimental model of  cardiomyocyte-specific knockout (KO) of MAO-B mice. They observed that MAO-B KO protects male mice against acute myocardial infarction, but had no effect on infarct size in female mice.

Although the manuscript deals with interesting topic it has serious flaws in several points that do not make it suitable for a publication in this form. The major point of criticism is that the manuscript is confused in the data presentation and analysis and not supported by an explicative discussion of the results.

Data and analyses are not presented appropriately. The comparison between/among groups are not clearly described.

The figures are often confused and without correct references to what is reported in the text.
They do not properly show the data and are not easy to interpret and understand.

There are some comments to be address:

1) In the results section, 2.1 paragraph, it is not clear if there is any significant difference in
mortality rate after I/R in MAO-B KO mice (as you indicated in lines 81-82). The following
sentences say something else.
What the phraseSummary of the number of operated mice before and after surgery” means in that context (lines 86-87)?

2) Results section, 2.2 paragraph. In figure 2 the indication of boxes A and B are missing.

Moreover, I suggest the authors to move the sentence referring to figure 2A (lines 97-98)
before the one describing figure 2B or, if they preferred, to change figure 2A and 2B.

The sentence contained from line 101 to line 103 should be moved to the materials and methods.
 How is EVAN Blue helpful for “area at risk” determination and how do you quantify infarct size with TTC?

3) Results section, 2.4 paragraph. Also for Figure 5 the indication of boxes A, B and C are missing.

Furthermore, the statistical analysis carried out is very confusing and the results of the comparison between/among
groups are not well described in the figure legend.

4) Result section, 2.5 paragraph. When were plasma 17β-estradiol levels measured in the experimental protocol? At what time? Why plasma 17β-estradiol levels is higher in male WT subjects compared female WT? And why didn't the authors measure the testosterone hormone? 

5) I believe the discussion needs to be edited to emphasize other differences in the regulation of cell survival pathways in males and females more than in the action of sex hormones.

Author Response

Reviewer 2

Comments and Suggestions for Authors

We thank the reviewer for her/his time and constructive critiques and suggestions to improve our manuscript. We have addressed each of the comments and have revised the manuscript accordingly.

There are some comments to be address:

1) In the results section, 2.1 paragraph, it is not clear if there is any significant difference in mortality rate after I/R in MAO-B KO mice (as you indicated in lines 81-82). The following sentences say something else. What the phrase“Summary of the number of operated mice before and after surgery” means in that context (lines 86-87)?

There was no difference in the mortality rate of male mice between WT and KO groups as well as between female WT and female KO groups. However, if we compared all males (including WT and KO animals) to all females (including WT and KO mice), we found a tendency for a decrease in mortality. In summary, sex difference slightly affected mortality but not the presence or absence of cardiac MAO B. We re-phrased the sentences of the lines 81-82 accordingly:

“There was no major difference in mortality rate between wild type (WT) and cardiomyocyte-specific MAO-B knockout (KO) females or males. However, comparing all male to female mice, there was a tendency for a higher mortality rate in males compared to females (p=0.103, when analyzed by Fisher’s exact test) (Figure 1).

The phrase “Summary….” is a part of the figure legend, thus, it is related to Figure 1, in which we demonstrate the numbers of all animals before I/R, number of animals that died after I/R, number of animals excluded as well as the number of surviving animals included in data analysis.

2) Results section, 2.2 paragraph. In figure 2 the indication of boxes A and B are missing. Moreover, I suggest the authors to move the sentence referring to figure 2A (lines 97-98) before the one describing figure 2B or, if they preferred, to change figure 2A and 2B.

Thank you for pointing this out. We have added the indication of boxes A and B and moved the sentence referring to figure 2A (lines 97-98) before the one describing figure 2B as suggested.

The sentence contained from line 101 to line 103 should be moved to the materials and methods.  How is EVAN Blue helpful for “area at risk” determination and how do you quantify infarct size with TTC?

Thank you for the comment and the question. The sentences of lines 101-103 have been removed from the figure legend of Figure 2, accordingly. The detailed description of the infarct size determination is provided in the Materials and methods section at 4.3. Determination of infarct size in vivo.

Briefly, Evans blue was injected in vivo into the apex of the left ventricle after re-occlusion of the coronary artery. Evans blue colors the whole left ventricle excluding the area supplied by the occluded coronary artery to dark blue, thereby demarcating the area at risk. In the next step, the heart is excised and cut into 5 to 6 slices and incubated in vitro with 1% TTC at 37°C. TTC binds to the reduced coenzymes such as NADH and FADH2 produced by the surviving cells, which leads to the conversion of TTC to a formazan dye and changes its color to red. Red-colored area is then identified as the surviving area inside the area at risk. Unstained, pale areas are identified as infarct size. For more details, please see PMID: 20188845 (doi: 10.1016/j.vascn.2010.02.014) for review.

Csonka C, Kupai K, Kocsis GF, Novák G, Fekete V, Bencsik P, Csont T, Ferdinandy P. Measurement of myocardial infarct size in preclinical studies. J Pharmacol Toxicol Methods. 2010 Mar-Apr;61(2):163-70. doi: 10.1016/j.vascn.2010.02.014. Epub 2010 Feb 25. PMID: 20188845.

3) Results section, 2.4 paragraph. Also for Figure 5 the indication of boxes A, B and C are missing. Furthermore, the statistical analysis carried out is very confusing and the results of the comparison between/among groups are not well described in the figure legend.

Thank you for pointing this out. We have added the indication of boxes A, B and C. We have changed the figure legend and assigned the comparisons between the groups to the respective figures as follows: Heart weight (HW) and body weight (BW) as well as HW/BW ratios of mice. Female mice had reduced heart weight (A) ***p<0.001 male vs female and ###p<0.001 WT male vs KO female, Students t-test; Two-way ANOVA: sex p<0.001. Female mice had reduced body weight (B) compared to male mice ***p<0.001 male vs female, ###p<0.001 WT male vs KO female, Students t-test; Two-way ANOVA: sex p<0.001. HW/BW ratios (C) were similar in WT males and females as well as in male cardiomyocyte-specific knockout (KO) mice, while in female KO mice HW/BW ratio was reduced *p<0.05 KO male vs KO female and ##p<0.01 WT male vs KO female, Students t-test. Two-way ANOVA: sex*genotype p=0.088. (WT male n = 14; WT female n = 9; KO male; n= 11; KO female n=11). Data are represented as box plots expressing median, 25% and 75 % quartiles, upper and lower whisker and outliers.

4) Result section, 2.5 paragraph. When were plasma 17β-estradiol levels measured in the experimental protocol? At what time? Why plasma 17β-estradiol levels is higher in male WT subjects compared female WT? And why didn't the authors measure the testosterone hormone? 

Plasma samples were taken at the end of reperfusion period (at 120 min) just before intracardiac injection of Evans blue.

The question of why 17β-estradiol is higher in males than in females after I/R is indeed of interest. Klaiber et al. (PMID: 7148879) reported that mean values of serum estradiol levels in men with myocardial infarction were significantly elevated over the comparable mean values of a respective control group. Estrogens can increase adrenergic activity, which would increase myocardial oxygen demand and testosterone can be converted to estradiol by aromatase in muscles.

Klaiber EL, Broverman DM, Haffajee CI, Hochman JS, Sacks GM, Dalen JE. Serum estrogen levels in men with acute myocardial infarction. Am J Med. 1982 Dec;73(6):872-81. doi: 10.1016/0002-9343(82)90779-3. PMID: 7148879.

5) I believe the discussion needs to be edited to emphasize other differences in the regulation of cell survival pathways in males and females more than in the action of sex hormones.

The reviewer is absolutely right. The importance of sex hormones in I/R damage cannot explain the observed findings as there is no indication in the literature that elevated estradiol levels in men are harmful (PMID: 24424037). However, it is undisputed that I/R injury differs between males and females and this might depend on the regulation of cell survival pathways. Myocardial I/R in females produces less cardiac damage, reflecting enhanced autophagy and decreased apoptosis (PMID: 24424037).

Le TY, Ashton AW, Mardini M, Stanton PG, Funder JW, Handelsman DJ, Mihailidou AS. Role of androgens in sex differences in cardiac damage during myocardial infarction. Endocrinology. 2014 Feb;155(2):568-75. doi: 10.1210/en.2013-1755. Epub 2013 Dec 11. PMID: 24424037.

Reviewer 3 Report

Sex difference in cardioprotection against acute myocardial infarction in MAO-B knockout mice in vivo

The article is interesting, the topic of myocardial infarction being always current.

There are other similar experimental studies that are centered on the same ideas as

Mitochondria from MAO-B KO hearts had a significant decrease in ROS production 

MAO-B KO hearts show no changes in ejection fraction and fractional shortening compared with WT hearts.

Cardiomyocytes-specific MAO-B KO decreased infarct size.

MAO-B KO lowered diastolic pressure after I/R and preserved post-ischemic contractile function.

The study design is well done as are the images. In addition, the influence of testosterone and estrogens on the gene expression of MAO, respectively on the size of the infarct area, was highlighted.

References must be up-dated

Heger J, Hirschhäuser C, Bornbaum J, Sydykov A, Dempfle A, Schneider A, Braun T, Schlüter KD, Schulz R. Cardiomyocytes-specific deletion of monoamine oxidase B reduces irreversible myocardial ischemia/reperfusion injury. Free Radic Biol Med. 2021 Mar;165:14-23. doi: 10.1016/j.freeradbiomed.2021.01.020. Epub 2021 Jan 18. PMID: 33476795.

Author Response

Reviewer 3

Comments and Suggestions for Authors

We thank the reviewer for her/his time and constructive critiques to improve our manuscript. We have addressed the comments and have revised the manuscript accordingly.

The article is interesting, the topic of myocardial infarction being always current.

There are other similar experimental studies that are centered on the same ideas as Mitochondria from MAO-B KO hearts had a significant decrease in ROS production 

MAO-B KO hearts show no changes in ejection fraction and fractional shortening compared with WT hearts.

Cardiomyocytes-specific MAO-B KO decreased infarct size.

MAO-B KO lowered diastolic pressure after I/R and preserved post-ischemic contractile function.

The study design is well done as are the images. In addition, the influence of testosterone and estrogens on the gene expression of MAO, respectively on the size of the infarct area, was highlighted.

References must be up-dated

Heger J, Hirschhäuser C, Bornbaum J, Sydykov A, Dempfle A, Schneider A, Braun T, Schlüter KD, Schulz R. Cardiomyocytes-specific deletion of monoamine oxidase B reduces irreversible myocardial ischemia/reperfusion injury. Free Radic Biol Med. 2021 Mar;165:14-23. doi: 10.1016/j.freeradbiomed.2021.01.020. Epub 2021 Jan 18. PMID: 33476795.

This manuscript is part of our reference list and is cited twice in the text.

Reviewer 4 Report

Please see attached PDF.

Author Response

Reviewer 4

We thank the reviewer for her/his time and constructive critiques and suggestions to improve our manuscript. We have addressed each of the comments and have revised the manuscript accordingly.

General Recommendations:

  1. Add sham data (if available) to Supplementary Table.
  2. Address the issue of Table 1 (similar estradiol).
  3. Add some general information to the Introduction and Discussion

Specific Comments:

Introduction:

(comment only, no response needed) The introduction is structured well but needs some work on introducing key concepts to understand the aim of the study. The potential differences between genders with regard to key hormones is mentioned only briefly and the effect of MAO inhibition on increases in catecholamines is not mentioned.

  1. Please insert more information about the terminal end of the MAO pathway regarding the action of FAD/FADH2 and increases in catecholamines and why that is clinically important.

We have added this information to the introduction as follows: The oxidative deamination of primary, secondary and tertiary amines is coupled to the reduction of the covalently linked co-factor flavin adenine dinucleotide (FAD) (Zhou et al., 1995).  With the aid of water and oxygen the formation of aldehydes, ammonia and the reactive oxygen species (ROS) molecule hydrogen peroxide (H2O2) are formed in the mitochondrial matrix and in the cytosol of the cell [2,3]Murphy 2009, Edmondson 2007; Youdim and Bakhle, BJP 2006. FAD is reduced to FADH2; it accepts two hydrogen atoms and undergoes a net gain of two electrons. H2O2 is generated by the recycling of FAD.The interaction of the FAD side chain with MAO-B is of utmost importance to incorporate FAD into MAO-B (Kirksey et al., 1998), since a deformation in the flavin ring negatively affects the reactive center and plays a crucial role for irreversible MAO inhibitor binding (Binda et 2002).  Accumulation of catecholamines can result in numerous diseases, such as the brunner syndrome (Van Rhijn et al., 2022), arrhythmia or in general cardiovascular problems (Schömig et al., 1990, Schömig et al., 1995).

Zhou, B.P.; Lewis, D.A.; Kwan, S.W.; Abell, C.W. Flavinylation of monoamine oxidase B. J Biol Chem. 1995, 270(40), 23653-60.

Youdim MB, Bakhle YS. Monoamine oxidase: isoforms and inhibitors in Parkinson's disease and depressive illness. Br J Pharmacol. 2006 Jan;147 Suppl 1(Suppl 1):S287-96. doi: 10.1038/sj.bjp.0706464. PMID: 16402116; PMCID: PMC1760741.

Kirksey, T.J.; Kwan, S.W.; Abell, C.W. Arginine-42 and Threonine-45 are required for FAD incorporation and caralytic activity in human monoamine oxidase B. Biochemistry 1998, 37, 12360-12366.

Binda, C.; Mattevi, A.; Edmondson, D. E. Structure-function relationships in flavoenzyme-dependent amine oxidations. J Biol Chem. 2002, 277, 23973-23976.

Schömig, A.; Richardt, G. Cardiac sympathetic activity in myocardial ischemia: release and effects of noradrenaline. Basic Res Cradiol. 1990, 85, Suppl 1:9-30.

Schömig, A.; Richardt, G.; Kurz, T. Sympatho-adrenergic activation of the ischemic myocardium and its arrhythmogenic impact. Herz. 1995, 20(3), 169-186.

  1. Please discuss briefly the increases in tyramine and serotonin that make MAO inhibition dangerous when taking multiple drugs. This allows the context of MAO-B inhibition to make more sense (since the MAO-B gives some cardiac benefit while MAO-A can maintain tyramine and serotonin homeostasis).

We have added this information to the introduction as follows: The MAO pattern in different cell types has been examined by use of specific inhibitors where MAO-A is inhibited by low concentrations of clorgyline and MAO-B by selegiline. Both isoforms are inhibited by pargyline (Kaludercic et al., 2014). MAOs play an essential role in the nervous system by modulating the levels of neurotransmitters. Various neurological diseases such as depression, Alzheimer's and Parkinson's are associated with altered MAO levels. The different preferences in substrate affinity are crucial for the different clinical significance of the two MAOs (Yeung et al., 2019). Inhibition of both isoforms or the use of irreversible MOA inhibitors can be fatal by the accumulation of MAO substrates such as tyramine or serotonin. Increased amounts of tyramine can enter the systemic circulation and, from there, adrenergic neurons, consequently increasing noradrenaline release resulting in severe hypertensive response (Finberg and Youdim 1985; Youdim and Bakhle, 2006). Similarly, an increase in serotonin concentration might be deleterious leading to serious adverse outcome including death (Scotton et al Int J. Tryp Res 2019). Selective MAO-B inhibition might provide a cardiac benefit while MAO-A can maintain tyramine and serotonin homeostasis.

Kaludercic N, Mialet-Perez J, Paolocci N, Parini A, Di Lisa F. Monoamine oxidases as sources of oxidants in the heart. J Mol Cell Cardiol. 2014 Aug;73:34-42. doi: 10.1016/j.yjmcc.2013.12.032. Epub 2014 Jan 9. PMID: 24412580; PMCID: PMC4048760.

Yeung AWK, Georgieva MG, Atanasov AG, Tzvetkov NT. Monoamine Oxidases (MAOs) as Privileged Molecular Targets in Neuroscience: Research Literature Analysis. Front Mol Neurosci. 2019 May 29;12:143. doi: 10.3389/fnmol.2019.00143. PMID: 31191248; PMCID: PMC6549493.

FINBERG, J.P. & YOUDIM, M.B. (1985). Modification of blood pressure and nictitating membrane response to sympathetic amines by selective monoamine oxidase inhibitors, types A and B, in the cat. Br. J. Pharmacol., 85, 541–546.

Youdim MB, Bakhle YS. Monoamine oxidase: isoforms and inhibitors in Parkinson's disease and depressive illness. Br J Pharmacol. 2006 Jan;147 Suppl 1(Suppl 1):S287-96. doi: 10.1038/sj.bjp.0706464. PMID: 16402116; PMCID: PMC1760741.

Scotton WJ, Hill LJ, Williams AC, Barnes NM. Serotonin Syndrome: Pathophysiology, Clinical Features, Management, and Potential Future Directions. Int J Tryptophan Res. 2019 Sep 9;12:1178646919873925. doi: 10.1177/1178646919873925. PMID: 31523132; PMCID: PMC6734608.

  1. MAO inhibitors have a long history of study as anti-depressants for recalcitrant depression and anxiety. There are notable sex differences in response to treatment that revolve around monoaminergic systems (dopamine, serotonin) and women tend to respond better to MAO inhibition than men. Although present in the Discussion, it might be better to mention this in the Introduction then tie in the hypothesis to this effect (estrogen vs. testosterone). By immediately linking the study to previous observations and then extending them to the heart, a strong link to previous literature can be established. In this way, in the Discussion, the short mention of depression and estrogen have proper context.

PMID: 28179816 might be useful for this.

Thank you for the citation. We have added this information to the introduction as follows: Women show a different susceptibility to diseases than men and react differently to treatment. The incidence of depression in women is nearly double than that in men. Female susceptibility may be linked to surges in reproductive hormones (PMID: 28179816). Women had a statistically superior response to MAOIs as antidepressant treatment (PMID: 12411218), demonstrated by elevated levels of serotonin, as well as the serotonin metabolite 5-hydroxyindoleacetic acid (5-HIAA), to be found in women compared to men (PMID: 28179816).  This makes a study of females and males with MAO-B KO even more promising.

Sramek JJ, Murphy MF, Cutler NR. Sex differences in the psychopharmacological treatment of depression. Dialogues Clin Neurosci. 2016 Dec;18(4):447-457. doi: 10.31887/DCNS.2016.18.4/ncutler. PMID: 28179816; PMCID: PMC5286730.

Quitkin FM, Stewart JW, McGrath PJ, Taylor BP, Tisminetzky MS, Petkova E, Chen Y, Ma G, Klein DF. Are there differences between women's and men's antidepressant responses? Am J Psychiatry. 2002 Nov;159(11):1848-54. doi: 10.1176/appi.ajp.159.11.1848. PMID: 12411218.

Methods:

  1. Although the IACUC most likely approved the protocol after demonstration of sample size calculations, it is best if a powers test or sample size calculations are in this section. Section 4.1 is a good place to add that very specific information (expected effect, number of animals required, etc.).

Thank you for the suggestion. We have included the data from the preliminary sample size calculation (https//:www.sample-size.net) into the manuscript. The values were setup to in case females: α=0.22 (type I error rate); β=0.44 (type II error rate); P0=0.15 (Group 0 risk); P1=0.4 (Group 1 risk); samples size=12; in case of males: α=0.22 (type I error rate); β=0.44 (type II error rate); P0=0.2 (Group 0 risk); P1=0.38 (Group 1 risk); samples size=24.

Results:

  1. (comment only, no response needed) Figures are easy to understand. Proper sizing and plotting make the data quickly comprehensible. Excellent work!

  1. The mortality rate for the male KO mice seems quite high. The WT mice have a post-IR mortality rate of about 25-29% but most of the female KO mice survived while the male KO mice experienced a post-IR mortality rate of about 43%. Is this a response to the surgery? Since there were no sham controls, the effect of the surgery

itself cannot be removed from analysis of the mortality. If this data is available, a Supplementary Table or Supplementary Figure would remove the surgery itself as a confounding factor.

Thank you for the comment. We did not use (and also did not plan to use) any sham control group, since our research group is well-experienced in mouse acute coronary occlusion surgery (see for publications PMIDs 29051737; 29769710; and 35742954), and based upon previous series of experiments, where the survival rate for sham animals was 100% and infarct size is close to 0%, we did not intend to use extra animals in line with the 2010/63/EU/ Directive on the care and use of laboratory animals in terms of the 3”R” rule, and due the limited availability of KO animals as well. Therefore, we cannot provide any sham data as supplementary material. In vast majority of the cases, mortality occurred during reperfusion due to acute heart failure.

  1. (comment only, no response needed) The inability of MAO-B KO to protect female hearts from reperfusion arrythmias is interesting information.

Arrhythmias are associated with calcium signaling. Estradiol is involved in calcium regulation and contractility in cardiomyocytes mediated by its receptors (PMID: 31156557).  A changed calcium balance could possibly contribute to the development of arrhythmias.

Mahmoodzadeh S, Dworatzek E. The Role of 17β-Estradiol and Estrogen Receptors in Regulation of Ca2+ Channels and Mitochondrial Function in Cardiomyocytes. Front Endocrinol (Lausanne). 2019 May 15;10:310. doi: 10.3389/fendo.2019.00310. PMID: 31156557; PMCID: PMC6529529.

  1. (comment only, no response needed) Table 1 is also interesting. If estradiol is protective against MI with respect to reperfusion injury in the KO females, the very similar levels of estradiol make the hypothesis difficult to support.

It is considered certain that the estradiol protects against cardiovascular disease in women (PMID: 21388771). A value for estradiol of 22-30 pg/mL seems to be protective (PMID: 22735686; PMID: 31156557), whereas values below but also above this value have been found to show the highest death rates from congestive heart failure (PMID: 19436016;). The levels of estradiol are almost similar in our mice after I/R, but the WT males show a slight increase. Interestingly, this is not unusual after I/R. Klaiber et al. (PMID: 7148879) reported that means of serum estradiol levels in men of a group with myocardial infarction were significantly elevated over the comparable mean of the control group. Estrogens can increase adrenergic activity, which would increase myocardial oxygen demand and testosterone can be converted to estradiol by aromatase in muscles.

Mosca L, Benjamin EJ, Berra K, Bezanson JL, Dolor RJ, Lloyd-Jones DM, Newby LK, Piña IL, Roger VL, Shaw LJ, Zhao D, Beckie TM, Bushnell C, D'Armiento J, Kris-Etherton PM, Fang J, Ganiats TG, Gomes AS, Gracia CR, Haan CK, Jackson EA, Judelson DR, Kelepouris E, Lavie CJ, Moore A, Nussmeier NA, Ofili E, Oparil S, Ouyang P, Pinn VW, Sherif K, Smith SC Jr, Sopko G, Chandra-Strobos N, Urbina EM, Vaccarino V, Wenger NK; American Heart Association. Effectiveness-based guidelines for the prevention of cardiovascular disease in women--2011 update: a guideline from the American Heart Association. J Am Coll Cardiol. 2011 Mar 22;57(12):1404-23. doi: 10.1016/j.jacc.2011.02.005. Erratum in: J Am Coll Cardiol. 2012 May 1;59(18):1663. PMID: 21388771; PMCID: PMC3124072.

Mahmoodzadeh S, Dworatzek E. The Role of 17β-Estradiol and Estrogen Receptors in Regulation of Ca2+ Channels and Mitochondrial Function in Cardiomyocytes. Front Endocrinol (Lausanne). 2019 May 15;10:310. doi: 10.3389/fendo.2019.00310. PMID: 31156557; PMCID: PMC6529529.

Jankowska EA, Rozentryt P, Ponikowska B, Hartmann O, Kustrzycka-Kratochwil D, Reczuch K, Nowak J, Borodulin-Nadzieja L, Polonski L, Banasiak W, Poole-Wilson PA, Anker SD, Ponikowski P. Circulating estradiol and mortality in men with systolic chronic heart failure. JAMA. 2009 May 13;301(18):1892-901. doi: 10.1001/jama.2009.639. PMID: 19436016.

Vandenplas G, De Bacquer D, Calders P, Fiers T, Kaufman JM, Ouwens DM, Ruige JB. Endogenous oestradiol and cardiovascular disease in healthy men: a systematic review and meta-analysis of prospective studies. Heart. 2012 Oct;98(20):1478-82. doi: 10.1136/heartjnl-2011-301587. Epub 2012 Jun 26. PMID: 22735686.

Klaiber EL, Broverman DM, Haffajee CI, Hochman JS, Sacks GM, Dalen JE. Serum estrogen levels in men with acute myocardial infarction. Am J Med. 1982 Dec;73(6):872-81. doi: 10.1016/0002-9343(82)90779-3. PMID: 7148879.

Discussion:

  1. If testosterone was previously found to be an unlikely mediator of this effect and estradiol was similar, the entire hypothesis was unsupported. Would it have been better to use a MAO-specific inhibitor in ALDH2 transgenic mice? Please address this possibility.

This is an interesting approach. There is a possibility of having an additive effect by the use of MAO inhibitors. If this is the case, MAO inhibition should also be able to confirm the gender difference.

We have added this information to the discussion as follows:

A possible approach to investigate the importance of gender, MAO-B and ALDH2 in the context of I/R injury would be the use of a MAO-B-specific inhibitor in ALDH2 overexpressing mice. The use of MAO inhibitors could lead to an additive effect thereby confirming the gender difference. If the cardioprotective effect in females is due to increased activity of ALDH2, no difference should be seen between ALDH2-overexpressing mice with and without MAO inhibition. However, these experiments would completely go beyond the scope of this work.

  1. IS is a non-standard abbreviation. Please spell it out on line 186.

We have changed it.

  1. Are male mouse hearts significantly larger than female hearts? If so, the infarct size theory in Line 205 is useful. But please address the issue of size first. In female humans, narrower trachea make asthma worse than in male humans so this possibility is plausible but the critical information of heart size is necessary to make that judgement.

Thank you for this comment. We can generally state that the size of female body (including height, weight and several other parameters) are smaller than that of male body in mice as well as in humans mostly due to the presence of testosterone and other androgenic hormones (anabolic steroids) in significantly higher concentration in males that in females. However, if we take heart weight/body weight ratio (as we have shown in the manuscript), we can conclude that there is no difference between wild type males and wild type females. Moreover, we expressed infarct size in the percentage of the occluded area and not in absolute values; therefore, the calculation of infarct size is independent from the size of the heart or the left ventricle. In case of trachea size and the severity of asthmatic symptoms, the reason for higher risk in females is probably due that the same degree of bronchoconstriction leads to a more severe obstruction with a smaller diameter trachea than that of with a larger caliber.

  1. (comment only, no response needed) Other factors, such as thromboxane A2 inhibition by estrogen and the effect of hormone replacement therapies on MAO activity make this paper clinically important.

That is probably true and very important and interesting aspect. However, these studies are far away from our actual field of research

Round 2

Reviewer 2 Report

Comment 1.

In the results section, 2.1 paragraph, it is not clear if there is any significant difference in mortality rate after I/R in MAO-B KO mice (as you indicated in lines 81-82). The following sentences say something else. What the phrase“Summary of the number of operated mice before and after surgery” means in that context (lines 86-87)?

Authors answer: OK      

The phrase “Summary….” is a part of the figure legend, thus, it is related to Figure 1, in which we demonstrate the numbers of all animals before I/R, number of animals that died after I/R, number of animals excluded as well as the number of surviving animals included in data analysis.

Authors answer: OK

Comment 2.

Results section, 2.2 paragraph. In figure 2 the indication of boxes A and B are missing. Moreover, I suggest the authors to move the sentence referring to figure 2A (lines 97-98) before the one describing figure 2B or, if they preferred, to change figure 2A and 2B.

Authors answer: OK

The sentence contained from line 101 to line 103 should be moved to the materials and methods.

Authors answer: OK

How is EVAN Blue helpful for “area at risk” determination?

Put the description “Briefly, Evans blue was injected in vivo into the apex of the left ventricle after re-occlusion of the coronary artery. Evans blue colors the whole left ventricle excluding the area supplied by the occluded coronary artery to dark blue, thereby demarcating the area at risk” in the Mat Met Section, 4.3. “Determination of infarct size in vivo”.

Comment 3.

Results section, 2.4 paragraph. Also for Figure 5 the indication of boxes A, B and C are missing. Furthermore, the statistical analysis carried out is very confusing and the results of the comparison between/among groups are not well described in the figure legend.

Authors answer: OK

Comment 4.

Result section, 2.5 paragraph. When were plasma 17β-estradiol levels measured in the experimental protocol? At what time?

Put the sentence “Plasma samples were taken at the end of reperfusion period (at 120 min) just before intracardiac injection of Evans blue” in the Mat Met Section, 4.4. Paragraph “Determination of 17 β estradiol

Why plasma 17β-estradiol levels is higher in male WT subjects compared female WT? And why didn't the authors measure the testosterone hormone?

I believe that you can add your considerations (Klaiber et al. reported that mean values of serum estradiol levels in men with myocardial infarction were significantly elevated over the comparable mean values of a respective control group. Estrogens can increase adrenergic activity, which would increase myocardial oxygen demand and testosterone can be converted to estradiol by aromatase in muscles) in the text to try to explain the phenomenon that you observed.

Comment 5.

I believe the discussion needs to be edited to emphasize other differences in the regulation of cell survival pathways in males and females more than in the action of sex hormones.

Authors answer: OK

Author Response

We thank the reviewer for her/his time and suggestions to further improve our manuscript. We have addressed each of the comments and have revised the manuscript accordingly.

Comment 2.

How is EVAN Blue helpful for “area at risk” determination?

Put the description “Briefly, Evans blue was injected in vivo into the apex of the left ventricle after re-occlusion of the coronary artery. Evans blue colors the whole left ventricle excluding the area supplied by the occluded coronary artery to dark blue, thereby demarcating the area at risk” in the Mat Met Section, 4.3. “Determination of infarct size in vivo”.

Answer:

We have added this paragraph to section 4.3. “Determination of infarct size in vivo“ (line 298-305).

Comment 4.

Result section, 2.5 paragraph. When were plasma 17β-estradiol levels measured in the experimental protocol? At what time?

Put the sentence “Plasma samples were taken at the end of reperfusion period (at 120 min) just before intracardiac injection of Evans blue” in the Mat Met Section, 4.4. Paragraph “Determination of 17 β estradiol”

Answer:

We have added this sentence to section 4.4. “Determination of 17 β estradiol” (line 338-339).

Why plasma 17β-estradiol levels is higher in male WT subjects compared female WT? And why didn't the authors measure the testosterone hormone?

I believe that you can add your considerations (Klaiber et al. reported that mean values of serum estradiol levels in men with myocardial infarction were significantly elevated over the comparable mean values of a respective control group. Estrogens can increase adrenergic activity, which would increase myocardial oxygen demand and testosterone can be converted to estradiol by aromatase in muscles) in the text to try to explain the phenomenon that you observed.

Answer:

We added our considerations concerning estradiol levels as suggested to the discussion part (line 218-222).

Reviewer 4 Report

All changes have been made and all questions answered.  

Author Response

We would like to thank the reviewer again very much for her/his time and constructive criticism and suggestions, which helped to improve our manuscript.